# Collaborative design of a decision aid for stroke survivors with multimorbidity: a qualitative study in the UK engaging key stakeholders

Talya Porat,[1] Iain J Marshall,[2] Euan Sadler,[3] Miguel A Vadillo,[4] Christopher McKevitt,[2] Charles D A Wolfe,[2] Vasa Curcin[2]

¹Dyson School of Design Engineering, Imperial College London, London, UK
²School of Population Health and Environmental Sciences, King's College London, London, UK
³School of Health Sciences, University of Southampton, UK
⁴Departamento de Psicología Básica, Universidad Autónoma de Madrid, Madrid, Spain

**Correspondence to**
Dr Talya Porat;
t.porat@imperial.ac.uk

## ABSTRACT

**Objectives** Effective secondary stroke prevention strategies are suboptimally used. Novel development of interventions to enable healthcare professionals and stroke survivors to manage risk factors for stroke recurrence are required. We sought to engage key stakeholders in the design and evaluation of an intervention informed by a learning health system approach, to improve risk factor management and secondary prevention for stroke survivors with multimorbidity.

**Design** Qualitative, including focus groups, semistructured interviews and usability evaluations. Data was audio recorded, transcribed and coded thematically.

**Participants** Stroke survivors, carers, health and social care professionals, commissioners, policymakers and researchers.

**Setting** Stroke survivors were recruited from the South London Stroke Register; health and social care professionals through South London general practices and King's College London (KCL) networks; carers, commissioners, policymakers and researchers through KCL networks.

**Results** 53 stakeholders in total participated in focus groups, interviews and usability evaluations. Thirty-seven participated in focus groups and interviews, including stroke survivors and carers (n=11), health and social care professionals (n=16), commissioners and policymakers (n=6) and researchers (n=4). Sixteen participated in usability evaluations, including stroke survivors (n=8) and general practitioners (GPs; n=8). Eight themes informed the collaborative design of DOTT (Deciding On Treatments Together), a decision aid integrated with the electronic health record system, to be used in primary care during clinical consultations between the healthcare professional and stroke survivor. DOTT aims to facilitate shared decision-making on personalised treatments leading to improved treatment adherence and risk control. DOTT was found acceptable and usable among stroke survivors and GPs during a series of evaluations.

**Conclusions** Adopting a user-centred data-driven design approach informed an intervention that is acceptable to users and has the potential to improve patient outcomes. A future feasibility study and subsequent clinical trial will provide evidence of the effectiveness of DOTT in reducing risk of stroke recurrence.

### Strengths and limitations of this study

► Engaging a range of stakeholders in the design and evaluation of an intervention ensures that the intervention is in line with the needs reported by the different stakeholders (eg, stroke survivors, healthcare professionals, policymakers).
► Adopting a learning health system approach enables the delivery of personalised recommendations in real-time while simultaneously capturing additional data back into the system, to improve the system's predictive model and recommendations.
► As only stroke survivors able to attend the focus groups participated in the study, we did not elicit the views of stroke survivors who are less mobile or housebound.

## INTRODUCTION

Stroke is the second leading cause of death and a major cause of disability worldwide.[1] In 2015, there were 3.7 million people living with stroke as a chronic condition in Europe and this number is expected to reach 4.6 million in 2035.[2] Stroke survivors have a ~40% cumulative risk of recurrence during the first 10 years after stroke.[3] Secondary stroke prevention requires healthcare professionals to offer effective interventions to monitor and manage risk factors, and for patients to change health-related behaviours (eg, smoking)[4] and adhere to preventative medications (eg, to control hypertension).[5] Follow-up appointments with clinicians offer opportunities to discuss interventions for reducing the risk of future stroke. However, long-term stroke care is characterised by a lack of continuity[6] and modifiable risk factors are currently not well detected, managed or controlled poststroke.[7]

Interventions designed to improve risk-factor management among stroke survivors in randomised controlled trials (RCTs) have shown modest or no effect. A recent

Cochrane systematic review of 42 RCTs evaluating the effectiveness of educational and behavioural or organisational interventions on modifiable risk factor control for secondary prevention of stroke, found no clear benefit in any of the target outcomes (ie, blood pressure, lipid profile, blood sugar, body mass index and recurrent cardiovascular events).[8] Possible reasons could be that these interventions have not been part of the clinical decision-making process of clinicians, did not engage various stakeholders in the design of the intervention and were not integrated with the electronic health record (EHR) (with the exception of one study[9])—all of which are considered critical features of successful clinical decision support systems (DSS).[10 11]

Stroke survivors commonly experience multimorbidity.[12] Gallacher and colleagues found that 94% of the people with stroke had one or more additional morbidities and often experienced long-term physical, psychological and social consequences.[12] This makes improving long-term stroke care a complex endeavour, requiring patient engagement, high-quality up-to-date information and a holistic approach which focuses on the patient and not on the disease.[13] These aspects are important both to plan effective treatments for individual patients and guide best practice for the stroke population in general.[14]

The learning health system (LHS) 'focusses on approaches to capture data from clinical encounters and other health-related events, analyse the data to generate new knowledge, and then apply this knowledge to continuously inform and improve health decision making and practice' (Nwaru, p177).[15] In a recent report (2019) stating what the NHS can learn from the LHS, the authors argue that it is necessary to use data to transform services, not just to digitise current ways of working.[16] Thus, LHS outputs can provide tailored information on optimal care decisions and be delivered at the point of clinical care.[17]

DSS which aim to analyse a patient's characteristics to provide tailored recommendations (such as for diagnosis,[18] treatment or long-term management), implement this transfer of evidence into practice. This is done particularly when used in conjunction with sources of 'real world data'[19] such as EHR systems that capture detailed data on specific conditions. Such point-of-care DSS support a range of applications, including identifying patient risk estimation, providing guidance on the appropriateness of treatments and tailoring clinical information to specific patient needs—providing the right care to the right patient at the right time.[17] A few studies have reported that engaging stakeholders to develop a LHS and integrated DSS improved patient outcomes and processes of care for individuals with long-term conditions.[20 21]

Increasingly, patients are expecting to be informed and involved in their care.[22] This shift from imposition of professional opinion towards a more collaborative model of care is not only relevant when people face difficult decisions about their health, where there are high stakes and where outcomes are uncertain, but also in situations where people need to manage long-term conditions or consider making changes in their lifestyles in order to reduce future risks.[23] Such shared decision-making (SDM) respects patient values and preferences, and supports decision-making through the provision of high-quality, accessible information.[24] SDM has been found to be most effective if interventions are developed for use during the clinical encounter,[25] and several DSS that have been designed to facilitate SDM during the consultation (ie, decision aids) have shown improved treatment adherence and clinical outcomes in patients with chronic conditions such as asthma and diabetes.[26 27]

In his seminal analysis, Berg criticised the 'top-down' technology centred approach to designing DSS.[28] He described an alternative *sociotechnical* approach, where new tools needed to be designed taking into account the real-world complex networks of people involved in healthcare, and designed using an iterative approach which makes strong use of qualitative research with users.

## Aims and objectives

The aim of this study was to engage key stakeholders to identify priorities and information needs in long-term stroke care and collaboratively design and evaluate a selected intervention that could be integrated as part of the EHR system, informed by a LHS approach. The data supporting the selected intervention are based on linked datasets from the South London Stroke Register (SLSR),[29] which includes >6000 records of first-ever strokes that occur in South London, and Lambeth Datanet (LDN)[30] containing primary care data of local general practices in South London.

## METHOD

### Patient and public involvement

The design was informed by active feedback from stroke survivors and carers from King's College London's (KCL's) Stroke Research Patient and Family Group (SRPFG),[31] a service user research group which consists of 32 participants currently on the SLSR who are from diverse socioeconomic and ethnic backgrounds. Stroke survivors, carers, health and social care professionals, commissioners, policymakers and researchers were involved throughout the study in a collaborative design and evaluation process.

### Data collection

We used a range of methods to engage stakeholders (n=53) in the design and evaluation of the intervention, including focus groups, face-to-face interviews and usability evaluations (see topic guides and interview questions in the online supplementary files). The process involved three main stages: (1) exploring stakeholder priorities for data and information needs to inform potential solutions for long-term stroke care; (2) collaborative design of the selected intervention with stakeholders, comprising cycles of design, prototyping and evaluation;

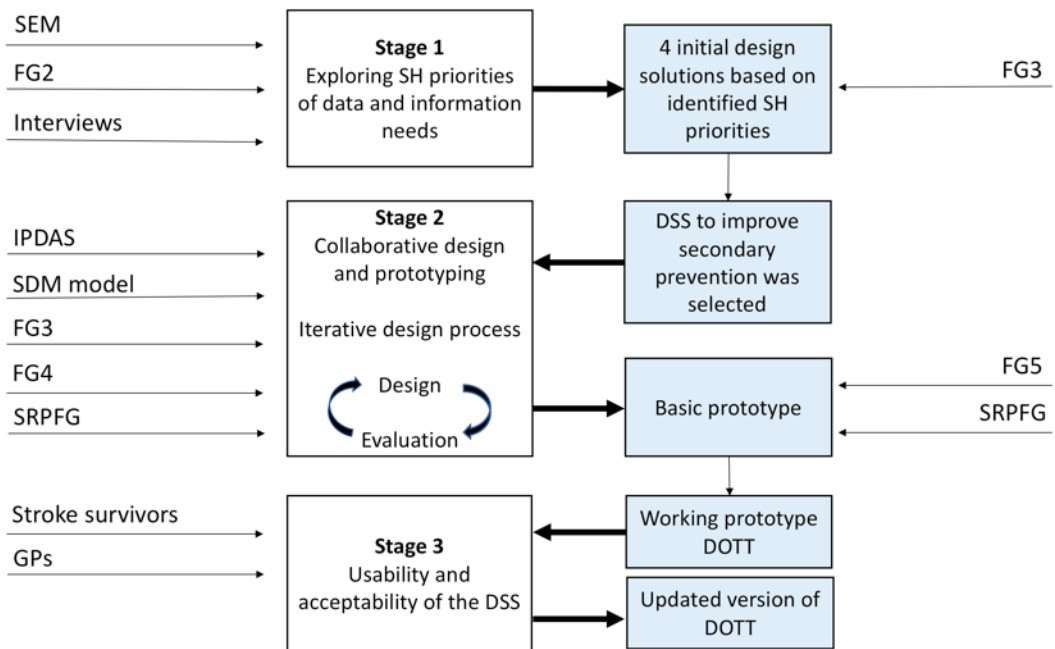

**Figure 1** A diagrammatic summary of the development and evaluation of DOTT, including the data that fed the different stages and the outputs. DOTT, deciding on treatments together; DSS, decision support systems; GPs, general practitioners; SEM, stakeholder engagement meeting (consisting three focus groups); FG, focus group; SH, stakeholders; IPDAS, International Patient Decision Aids Standards[23]; SDM model, shared decision-making model for clinical practice[32]; SRPFG, Stroke Research Patient and Family Group.[31]

(3) usability and acceptability evaluation of the DSS prototype (see figure 1). Thirty-seven stakeholders participated in the first two stages, including stroke survivors and carers (n=11), health and social care professionals (n=16), commissioners and policymakers (n=6) and researchers (n=4). Sixteen stakeholders participated in the third stage, including eight stroke survivors and eight general practitioners (GPs). Stroke survivors were recruited from the SLSR. Health and social care professionals were recruited through general practices in South London and KCL networks. Carers, commissioners, policymakers and researchers were also recruited through these networks. Stakeholders were purposively sampled to include stroke survivors (ie, men and women, with a range of disabilities and long-term conditions, risk factors and length of time since their stroke) and professionals providing all types of stroke care and support. See table 1 for details of all stakeholders taking part in the study. Participants could take part in the study if they were able to attend the meetings and were willing to sign a consent form. Transport was arranged for less mobile patients.

### Stage 1: exploring stakeholder priorities for data and information needs

In total, 37 stakeholders participated in this stage. An initial stakeholder engagement meeting (SEM) comprising 24 participants, 9 face-to-face interviews with key stakeholders who could not attend this meeting, and a second focus group involving 12 participants (FG2) were conducted (some participants took part on multiple occasions). The methods and findings from this stage

of the study have been reported elsewhere.[14] In brief, in the initial engagement meeting (SEM), participants were introduced to the concept of a LHS and then in three separate focus groups (service user/carer; health and social care professionals; commissioners and policymakers) they were asked to identify priorities and potential solutions that may be derived from the clinical data to improve long-term stroke care for stroke survivors with multimorbidity. Then, in the larger group, through a process of priority setting and consensus led by a facilitator (ES), stakeholders identified a number of priorities and solutions to improve long-term management of stroke (ie, improving continuity of care; improving management of mental health consequences; better access to health and social care; and targeting multiple risk factors). Interviews with clinicians who could not participate in the meetings also took place, to ascertain their views on priorities and potential solutions using clinical data. A core stakeholder group was then established to work collaboratively with the research team to design potential interventions and to provide their active feedback (FG2). This core stakeholder group (n=12) comprised stroke survivors, healthcare professionals, carer, policymaker and commissioner.

Targeting multiple risk factors after stroke was identified among stakeholders as a key priority, and a DSS to improve secondary prevention after stroke to target multiple risk factors was subsequently chosen within the core stakeholder group (FG3) for further development (n=10).

**Table 1** Stakeholders taking part in the study

| Type of stakeholder | SEM (n=24) | Interviews (n=9) | FG2 (n=12) | FG3 (n=10) | FG4 (n=9) | FG5 (n=9) | Usability evaluation (n=16) | Total (n=53*) |
|---|---|---|---|---|---|---|---|---|
| **Stroke survivor** | 10 | | 2 | 2 | 2 | 2 | 8 | 18 |
| **Carer** | 1 | | 1 | 1 | 1 | 1 | | 1 |
| **Health and social care professional** | 8 | 7 | 3 | 2 | 2 | 2 | 8 | 22 |
| GP | 2 | 5 | 1 | 1 | 1 | 1 | 8 | 13 |
| Physiotherapist | 2 | | 1 | | | | | 2 |
| Speech and language therapist | 1 | | | | | | | 1 |
| Social care professional | 1 | | | | | | | 1 |
| Public health doctor | 1 | | | | | | | 1 |
| Consultant psychiatrist | 1 | | | | | | | 1 |
| Occupational therapist | | | 1 | 1 | 1 | 1 | | 1 |
| Acute stroke care consultant | | 2 | | | | | | 2 |
| **Policymakers and commissioners** | 3 | 2 | 2 | 2 | 2 | 2 | | 6 |
| **Third sector representatives** | 2 | | | | | | | 2 |
| **Academic researchers** (social scientist, researchers working with SLSR/LDN databases) | | | 4 | 3 | 2 | 2 | | 4 |

King's College London's Stroke Research Patient and Family Group (SRPFG) comprising 32 stroke survivors and carers also provided feedback on the design of the intervention in two of their meetings.
*Overall 53 participants took part in the study, but a number of stakeholders took part on multiple occasions.
The bold fonts highlight the row that summarises the number of participants (eg, the row 'Health and social care professional' summarises the rows below (eg., GP, physiotherapist, etc).
FG, focus group; GP, general practitioner; LDN, Lambeth Datanet; SEM, stakeholder engagement meeting; SLSR, South London Stroke Register.

## Stage 2: collaborative design and prototyping of selected intervention

The initial design of the DSS to improve secondary stroke prevention and target multiple risk factors after stroke was informed by the first stage and guided by the International Patient Decision Aids Standards (IPDAS),[23] which provides a framework and standards for the design of patient decision aids, and the SDM model for clinical practice.[32] The latter provides a model of how to conduct SDM in practice based on providing patients choice, a range of options and involving them in 'decision talk'. Following feedback from the core stakeholder group at the third focus group meeting above (FG3), an updated design of the intervention was subsequently reviewed by the core stakeholder group at a fourth focus group (n=9) (FG4) and was revised following their feedback. The DSS was also presented to the KCL's SRPFG. The intervention was revised and the updated design was developed as a basic prototype and was further discussed during a subsequent focus group with the core stakeholder group (n=9) (FG5) and the SRPFG. This process allowed all stakeholders to iteratively develop and refine the DSS to a working prototype.

## Stage 3: usability and acceptability evaluation of the DSS

Sixteen participants, including eight stroke survivors and eight GPs participated in the usability and acceptability evaluation of the working prototype of the DSS. None had taken part in the previous stages of the study.

The evaluation included simulated consultations using the DSS prototype. In the GPs session, the researcher acted as the patient, and in the stroke patient's session, the researcher acted as the GP. GPs were given a short tutorial on how to use the DSS before the simulated consultations and stroke survivors were given a short explanation about the DSS. GPs and stroke survivors were interviewed after the simulated consultation, asking them to provide feedback on the DSS, including its strengths, limitations and suggestions for improvements. Stroke survivors and GPs also answered an acceptability questionnaire[33] and the System Usability Scale (SUS).[34] Acceptability relates to the comprehensibility of the components of the decision aid, including its length, pace, amount of information, balance in presentation and overall suitability.[33] Usability is 'the extent to which a product can be used by specified users to achieve specified goals with effectiveness, efficiency, and satisfaction in a specified context of use'.[35] The SUS is composed of 10 questions and has been shown to be a reliable and psychometrically validated tool.[36] Ratings were provided on 5-point Likert scales from 1 (strongly disagree) to 5 (strongly agree), with higher ratings indicating higher satisfaction.

For the usability evaluation, the DSS prototype had the following functionality and flow:

1. Stroke survivors (patients) indicated their perceived risk of having a recurrent stroke.
2. GPs entered the patient's characteristics (age, gender, clinical conditions).
3. The system displayed a 'typical' recurrent stroke risk (age group-specific average)[37] and the most effective treatments based on the patient's characteristics.
4. The benefit of each treatment in terms of reducing the stroke risk was displayed. Estimated relative stroke risk reductions were calculated based on the existing literature.[38–41]
5. Information and common concerns for each treatment were displayed.
6. The GP and patient decided on a management plan while identifying desired clinical and patient outcomes.
7. Patients were told that their management plan would be printed to take home.

### Data analysis
Data from focus groups and interviews were audio recorded, transcribed in full and stored in NVivo V.11. Qualitative data were analysed using a thematic analysis approach[42] to identify themes and subthemes related to stakeholder perspectives informing the identification, design and evaluation of a DSS to improve secondary prevention for stroke survivors, which could be part of a LHS. This involved two authors (TP, ES) assigning codes and refining themes from the data, noting similarities and differences between stakeholder perspectives. The two authors have doctoral/post-doctoral experience in conducting and analysing qualitative data in applied health research.

### RESULTS
#### Focus groups and interviews
Eight themes related to improving secondary prevention and management of multiple risk factors after stroke were identified from focus groups and interviews.

#### Theme 1: Involve stroke survivors in decisions concerning their treatments
In the focus groups, stroke survivors often articulated that due to their multiple health conditions, and hence multiple risk factors for stroke recurrence, they would like to be more involved in selecting their treatments based on what is important to them and their desired outcomes. This viewpoint was further confirmed by stroke survivors participating in KCL's SRPFG. A number of clinicians perceived that SDM did not take place on a regular basis during routine clinical consultations, and there was a need for greater involvement of stroke survivors and their carers in selecting treatments that best meet their needs and preferences. Commissioners and policymakers agreed that SDM is a necessity and noted that policies in the UK and other countries required the involvement of patients in their treatment decisions. They also emphasised the importance of data and evidence-based recommendations to improve decision-making about treatments.

> When I go to my doctor I realise it's my doctor who is making the decisions…but I think that patients now know often more about their own condition than the health professionals (stroke survivor, SEM).

> This information (risk factors) which used to be something that I, as a doctor, only thought about, it's now something that we should think about together (GP, FG5).

> How do we help patients and carers and health professionals together have a discussion using data information to make decisions about treatments? (commissioner, FG2).

#### Theme 2: Present and communicate recurrent stroke risk in a meaningful way
Both stroke survivors and healthcare professionals (in the focus groups and interviews) emphasised the importance of displaying and communicating personalised stroke risk estimation in a clear and meaningful way. Stroke survivors expressed that current risk presentations lacked clarity, with healthcare professionals agreeing with this idea, reporting that they also find it difficult to understand and communicate risk to patients while linking it to specific actions and behaviours among patients.

> What is this individual's risk of a further stroke in five years… and that's really important because patients commonly ask us that 'what is the risk of me having another stroke in the next year' and we come up with a figure and we say '5% of whatever' (hospital stroke physician, Interview).

> And I think the other thing is what actually is risk, how do you convey that, I mean, is it twice as much risk if I've never had a stroke…I know exactly what you mean 50% and 5% of that are meaningless to most people (stroke survivor, FG4).

> Because the patients often think that the GPs—or the doctors/the specialists understand risk. It's really difficult to understand risk and we have to use guidelines to help us with risk. So if the guidelines say, 'This is a risk and this is the level at which you should intervene', then I'm not well enough informed to go any further than that (GP, FG3).

#### Theme 3: Compare stroke survivor's perceived stroke risk with their predicted risk
In one of the focus groups, a carer voiced the importance of allowing stroke survivors to articulate their own perceived risk of having a recurrent stroke, which could then be compared with the actual predicted risk. Professionals and lay stakeholders in the group agreed that this would facilitate a collaborative discussion on potential risk factors and their impact on stroke risk.

 

Patients themselves if they've been through a process will likely at some point be shown something and said either mark yourself on this, because another thing is where do you think you are on this scale at the moment with your risks, sometimes that's quite powerful (carer, FG4).

### Theme 4: Personalise treatments to help control multiple stroke risk factors

Different stakeholders in a number of the focus groups and interviews emphasised the importance of controlling multiple risk factors for stroke recurrence in stroke survivors with multimorbidity and the need to develop effective treatments based specifically on the patient's characteristics (eg, age, ethnicity, health conditions). Stroke survivors from the SRPFG similarly voiced their preference to know their personal risk according to their personal characteristics and receive tailored advice from professionals about what specific actions they could perform to reduce the identified risks. Commissioners were interested in care pathways for stroke patients with multimorbidity and how these care pathways could be tailored to the patient's characteristics.

Patients who've had a confirmed stroke, the first thing as a family physician in terms of management is to make sure that you've controlled all their risk factors to prevent them getting another stroke (GP, Interview).

And if the system could provide him, like, tailored for the patient taking all the information and saying OK for this patient because he had stroke, he has diabetes and high blood pressure, we recommend the following care pathway, treatments (commissioner, SEM).

Anything that can be personalised or tailored, so you don't feel it's this off the shelf thing that you're being given, you know… you sit with your doctor and it's not just a case of giving out a leaflet, but actually let's have a look at your personal data (occupational therapist, FG4).

### Theme 5: Display effectiveness of recommended treatments in reducing stroke risk

The majority of health and social care professionals, commissioners and policymakers perceived that stroke survivors with multimorbidity often have multiple risk factors to manage, and that prioritising the different treatments available for secondary prevention of these risk factors was required. Stroke survivors wanted to know the relative benefit of the proposed treatments being offered by clinicians in terms of how they addressed stroke risks and to take this information into account when deciding on personalised treatments. Commissioners specifically emphasised the importance of using evidence-based data to prioritise treatments to help patients in their decision-making.

…and you need to know, in fact, what the risk is if you do nothing compared with the risk if you do something (stroke survivor, FG3).

The question might be for a patient 'should I take a statin after a stroke' and we might be able to use the database to answer the question 'what would be the risk of future stroke if I do take a statin or if I don't take a statin' and you can use that information to help to come to a decision together (commissioner, SEM).

Well I suppose you could think about the common comorbidities, so hypertension and stroke, AF (atrial fibrillation) and stroke, diabetes and stroke and you could think about not necessarily an algorithm but a sort of stepwise prioritisation about what you should think about in terms of the patient's total management, you know, which would be the most important area of focus? (GP, Interview).

### Theme 6: Address stroke survivor concerns about treatment and barriers to adherence

Stroke survivors in some of the focus groups and the members of the SRPFG raised concerns about the challenges of multiple treatments they were expected to adhere to in order to decrease the potential risks of a recurrent stroke, commonly reporting that they did not always understand the value of these treatments. Several felt that a joint discussion with a healthcare professional about these concerns would help them better understand the value of a particular treatment and reach an informed decision about it. When interviewed, several GPs agreed that it was very challenging for stroke survivors with multimorbidity to adhere to multiple medications and other treatments at any given time, and that it is sometimes difficult to identify among their various treatments what is absolutely necessary and what is 'good to have'.

My experience both with the doctors at the surgery and the consulting hospital is trying to discuss the medication that they insisted I took. I had horrendous side-effects and I kept trying to say to them 'Look, I'm having these side-effects, can I change, can I reduce, can I do blah blah' and their attitude I have to say, is one of terrorising patients (stroke survivor, SEM).

I think that's a common problem with all patients that suffer from comorbidities. It's rationalising their medication and you know being able to take a holistic view of the person and make sensible decisions about what they absolutely need to continue on and what they don't. And you can only really do that just by having time with the patient, you know if it's important for them to be able to sort of get up and get out and about and not feel dizzy, then you may have to compromise on how much blood pressure medication they take (GP, Interview).

### Theme 7: Support continuity of care

Stroke survivors commonly reported that they do not have appointments with their GP or other healthcare

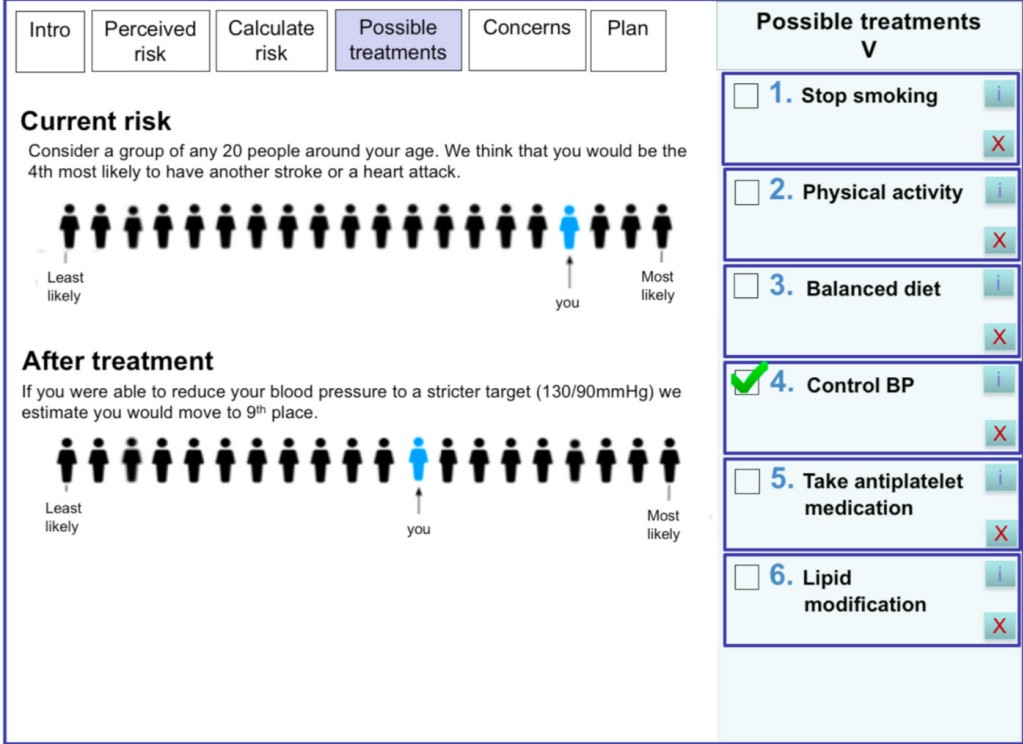

**Figure 2** An example screen from DOTT prototype displaying the stroke survivor's predicted stroke risk before and after a selected treatment (eg, control of blood pressure). DOTT, deciding on treatments together.

professionals on a regular basis. Several felt that the idea of personalised care to control stroke risk factors is very important but should have a follow-up to ensure continuity of care, which was often lacking. Some also perceived that the selected treatments and management plan should be saved on the system for future consultations and a follow-up appointment always set in advance. Commissioners also emphasised the importance of follow-up appointments and raised the concern that although follow-up appointments are an important part of stroke management and are required according to the National Institute for Health and Care Excellence (NICE) guidelines, many stroke survivors do not have follow-up appointments and do not see a GP over the longer term.

> I'm just thinking of my practice where it's very difficult to get to see the same doctor and if I was presented with my third in line (ie, the risk graphic display) ten times from ten different doctors I'd be starting to get a bit hacked off I think (stroke survivor, FG4).

> It's not a one time thing…there needs to be continuous interaction I think if something's going to happen (stroke survivor, FG4).

### Theme 8: Identify stroke survivors at high risk of recurrent stroke

Healthcare professionals, commissioners and policymakers highlighted the need to proactively identify stroke survivors at high risk of having a recurrent stroke to assess and treat them in a timely manner. They felt that many stroke survivors, especially those with more severe long-term consequences from the stroke, do not often see a

physician, and it is important to have a smart (automatic) system in place that could proactively identify them and assess their risks.

> I think the challenge first of all who are the high-risk patients, can we identify them and, if we can, is there a way through case management or community matrons, you know, linked with the stroke teams in the community providing access to therapy and assessment when it's required in a timely fashion (commissioner, Interview).

### Development of DOTT decision aid

The above themes and solutions were proposed, designed and refined during the collaborative design process with stakeholders, which informed the design of DOTT (Deciding On Treatments Together). DOTT is a computerised decision aid (ie, a DSS designed to facilitate SDM), integrated with the EHR system, to be used in primary care during clinical consultations between the healthcare professional and stroke survivor, aiming to facilitate SDM on treatments to reduce recurrent stroke risk.

Specifically, DOTT will:

1. Allow stroke survivors to indicate, in a graphic presentation (figure 2), *their perceived risk of having a further stroke*. The graphic presentation in DOTT is based on population rank[43 44] simulating a queue of 20 people around the same age of the stroke survivor. Stroke survivors indicate where they think they are positioned in the queue (from least to most likely). This risk would

then be compared with the actual predicted risk to facilitate conversation on risk factors. Needs from theme 3 are addressed with this feature.

2. Display *stroke survivor's predicted risk of having a further stroke* in a meaningful and understandable way for both healthcare professionals and stroke survivors. For the usability evaluation, the system displayed a typical recurrent stroke risk based on age.[37] The final personalised stroke risk model is under development and will be calculated based on the patient's information from the EHR and on rules generated from the linked dataset (SLSR and LDN). This will include variables such as age, gender, medical history (eg, hypertension, atrial fibrillation), type of stroke and time since stroke. Needs from theme 2 are addressed with this feature (see figure 2).

3. Provide a *list of personalised recommended treatments* for stroke survivors based on their risk factors (eg, hypertension, atrial fibrillation) extracted from the EHR. A list of the most effective evidence-based treatments for secondary prevention would be compiled and extracted from the recent NICE guidelines[45] and the National Clinical Guideline for Stroke.[46] This includes both clinical and lifestyle recommendations. For each recommended treatment, the evidence supporting the treatment will also be displayed. Needs from theme 4 are addressed with this feature.

4. *Prioritise the recommended treatments* based on their relative risk reduction and present the most effective treatment first. The clinician and stroke survivor can select one or more treatments and see on the graphic display, how the treatments reduce the overall stroke risk. The benefit of each treatment in terms of stroke risk will be calculated using the linked dataset (SLSR and LDN). Needs from theme 5 are addressed with this feature.

5. *Display stroke survivors' common concerns* on the suggested treatments (eg, 'do I have to take blood pressure drugs for life?'), which will aid in identifying and addressing barriers to treatment adherence and eliciting preferences. An initial list of concerns and their response was prepared based on qualitative studies eliciting patients' barriers to treatment adherence.[47 48] Needs from theme 6 are addressed with this feature.

6. Allow stroke survivors and their carers to discuss the different treatments with the healthcare professional and *jointly select the treatments that best suit the stroke survivor's preferences, desired outcomes and goals* (and remove the ones that do not). Lifestyle modification will be discussed during the consultation and enhanced through referral to specialists or lifestyle intervention programmes. The agreed management plan and information on the different treatments will be printed and handed to the stroke survivor to take home. Needs from theme 1 are addressed with this feature.

7. Set automatically a *follow-up appointment* in 3 months' time. The information entered, including the agreed management plan is saved and transferred back to the stroke survivor's EHR for future consultations.

During the follow-up consultation, the management plan is reviewed and treatments to address risk factors for stroke recurrence can be added, modified or removed. Desired clinical and patient outcomes will also be reviewed. Current NICE guidelines[45] for 'Secondary prevention following stroke and TIA' recommend primary care follow-up on discharge, 6 months and then annually. A 3-month follow-up was selected as a reasonable interval for healthcare professionals and to provide enough time for patients to adhere to the selected treatments. Needs from theme 7 are addressed with this feature.

8. The stroke prediction model will also be used to *proactively identify individuals at high risk of a recurrent stroke* by calculating their recurrent stroke risk at defined periods of time (the practice can define the desired threshold) and alert the practice (eg, physician, nurse, receptionist) to invite those patients for a clinical consultation. Needs from theme 8 are addressed with this feature.

9. All information from patients and healthcare professionals (eg, treatments selected by the patient, desired outcomes, predicted stroke risk, results in follow-up) will be *captured by the system as part of a LHS* and be used to improve the system's predictive model and treatment recommendations.

## Usability and acceptability evaluation
### Demographics
Eight stroke survivors and eight GPs participated in the usability and acceptability evaluations. GPs (four men, four women) had average of 10.3 years of experience as a GP. All had experience in providing care to stroke survivors, had medium to high confidence in using new technology and low to medium experience using DSS. Stroke survivors (four men, four women) had an average age of 65.5 years (SD: 11.4, range: 49–81). All had hypertension, two had heart problems, one was suffering from depression, four had mobility issues and four had minor cognitive deficiencies (attention and memory).

### Usability and acceptability
Both GPs and stroke survivors found the decision aid usable and acceptable. GPs found the decision aid easy to use (score 4.3), easy to understand (4.1) and felt very confident using it (4.2). They thought that this decision aid was better than how they usually helped patients decide about treatments for controlling their risk factors (4.4), that this strategy was compatible with the way they thought things should be done (4.3), that this type of decision aid was suitable for helping patients make informed choices (4.0) and that the decision aid complemented their usual approach (4.4). Stroke survivors perceived that they would like to use the decision aid frequently (4.0), thought that it was easy to use (4.2) and felt confident using it (4.1). Initial findings of the usability evaluation can be found in Porat *et al*.[49]

## Identified themes

Seven main issues relating to the usability and acceptability of the decision aid were identified. These were divided into themes relating to the importance of the decision aid, its functionality and concerns from using it.

### Importance of the decision aid
#### *Logical and structured process that facilitates discussion*

All GPs and stroke survivors (n=16) found the decision aid to be clear, and consisting of a logical flow that helped to structure the consultation. They felt that the decision aid facilitated a transparent discussion on the different proposed treatments and elicited patients' preferences.

> Physician pointing out what to do but the patient makes the decision since it's hard to get your head around everything. More doable if you have specific areas to work on with specific targets that suits you (stroke survivor 2).

#### *Importance of a learning system*

Several GPs (n=3) raised the importance of a learning system providing up-to-date information. They wanted to make sure that the suggested treatments are in line with the most up-to-date evidence.

> The learning aspect is very important, since this system is based on evidence and evidence can change (GP 6).

#### *Can motivate patients to change behaviour*

All GPs and stroke survivors (n=16) believed that the decision aid could motivate patients to change behaviour (eg, take their medication to reduce blood pressure, increase physical activity, eat healthy). Stroke survivors liked the idea of being involved in deciding on their treatments according to their preferences and abilities, receiving information on their stroke risk factors and discussing their views and concerns with their GP. They felt it gave them more control over their health and motivation to adhere to the treatments they selected. GPs felt it was a good way to discuss the different treatments and give patients the power to decide on treatments that suit them. A number of GPs and stroke survivors agreed that sharing decisions and enabling patients to select the treatments that best meet their preferences and goals, may increase patients' feeling of ownership over their health and improve adherence to the selected treatments.

> I believe discussing the different options with the patients, shared decision making, is likely to improve adherence (GP 1).

### Functionality
#### *Powerful risk display showing the benefit of each treatment*

The vast majority of GPs and stroke survivors (n=15) found the visual display showing the risk before and after a selected intervention, easy to understand, with some viewing it as a 'powerful' tool. Both stroke survivors and GPs commented that they were not aware of the effect the treatments have on reducing the stroke risk.

> The most powerful thing is the visual shifting of risk (GP 5).

> Wow, a small change can make a big difference, this is very encouraging (stroke survivor 6).

#### *The patient takes home printed information*

GPs and stroke survivors (n=10) thought that it was very important that the patient has a copy of the management plan and all the information printed so they can review it at home. In particular, stroke survivors wanted to have their current predicted risk and information on their selected treatments, including the date of the follow-up appointment printed out, so it could motivate them to adhere to their treatments.

> The important thing is that the patient goes out with a piece of paper that summarises in bullet points the outcome of the consultation. If its black and white on paper it makes a difference (stroke survivor 3).

### Concerns

GPs and stroke survivors raised two main concerns from using the decision aid.

#### *Deals with one aspect of the consultation*

GPs and stroke survivors (n=6) felt that the decision aid is good but focuses on one aspect of the consultation (reducing risk of recurrent stroke) and patients may have other concerns, such as depression or social isolation.

> This is good, but for me the most important thing is the emotional aspect, and this tool doesn't relate to that (stroke survivor 4).

### Time

The main concern for GPs was time (n=6), in which within the allotted standard 10 min for the consultation already provided significant limits, and most felt they will not manage to fit it in.

### Suggestions for improvement

GPs and stroke survivors provided suggestions for improving the decision aid:
1. The terminology was too clinical, for example, 'treatments' and 'management', could be changed to 'possible strategies or approaches'.
2. In addition to the management plan, information (eg, in the form of a leaflet) on each of the selected treatments should also be printed out and given to patients.
3. Add clinical data, for example, when clicking on 'cholesterol' show the patient's last three values, and do this also for their blood pressure.
4. Enable more than one display of risk, because each patient may prefer a different display and understands risk differently.

5. Add emotional and mental health aspects which are related to stroke risk.

We subsequently made the above changes and additions to the updated version of DOTT.

## DISCUSSION

Our work focused on engaging various stakeholders in the identification, design, prototyping and evaluation of a decision aid to improve secondary prevention after stroke. Eight themes informed the design of DOTT. A number of the themes and solutions proposed by the stakeholders have been implemented previously to some extent to support other patient groups, such as diabetes and atrial fibrillation.[50 51] These include, predicting a patient's risk based on their risk factors, proposing possible treatments and displaying their benefit in decreasing the risk[50] and incorporating patients' concerns within the decision-making process.[51] These themes were found useful and are recommended in SDM tools (eg, in the IPDAS).[23]

Additional unique themes and solutions have emerged as outcomes of the collaborative design process in this study, which could be used for a range of chronic diseases requiring long-term management. Specifically:

### Present and communicate risk in a meaningful way

While there are many different ways to communicate multiple risks to patients, the most commonly used are absolute or relative risks presented as percentages or probabilities (eg, 'from 100 people like you 20 are expected to have a recurrent stroke').[52] However, studies have shown that in general, healthcare professionals are as unfamiliar as their patients with risk estimates and probabilities[53] and often find it difficult to combine multiple risk factors into an accurate assessment of vascular risk[54] and to communicate this risk to patients.[55] Moreover, patients may feel that statistical risk estimates do not apply to them personally.[56] To overcome this, our graphic presentation is based on population rank, simulating the patient in a queue of people around their age.[43 44] Studies have also shown that formats which present data framed as the risk of an individual were perceived as more relevant and easier to relate to than percentage risk estimates.[57]

### Compare patient's perceived risk with their predicted risk

This is a novel requirement from a DSS, which to our knowledge does not exist in current systems. Perceived risk of adverse outcomes such as stroke may be an important concept in understanding patient's adherence to medication and recommended health behaviours.[58] Overall, patients tend to underestimate their own risk.[59] This tendency was also found when patients estimated their cardiovascular risk.[60] Weinstein refers to this underestimation as an 'optimistic bias'.[59] For example, a recent study found that people with undiagnosed diabetes or pre-diabetes considerably underestimated their probability to have or develop diabetes.[61] Lower perceived risk has been associated with poorer adherence to recommended health behaviours[62] and hence a more realistic perception of risk may increase patients' interest in risk reduction.[62] Research has shown that individualised risk feedback was effective in increasing perceived stroke risk among patients who had underestimated their stroke risk at baseline.[63] This may imply that eliciting patients' perceived risk and showing them the actual predicted risk, can change their inaccurate risk perception and increase their interest in risk reduction.

### Prioritising treatments

Healthcare professionals have previously expressed concerns about managing care and making decisions about treatments, including communicating risks and benefits for patients with multimorbidity and complex needs.[64] They commonly report having to make decisions with such patients which involve a process of prioritisation or trade-offs, facilitating a discussion with the patient on what is important to the patient and what they would like to achieve in terms of their health (ie, goal setting).[64] Aligning patient goals and desired outcomes with clinicians' goals is likely to improve outcomes for these patients.[65]

### Identify individuals at high risk

Calculating periodically (in an automatic way) the stroke risk of survivors to identify individuals at high risk of recurrent stroke (based on their information in the EHR) could be a valuable feature for improving long-term management and care for stroke survivors who are less likely or able to visit healthcare professionals on a regular basis. This theme was identified and prioritised by healthcare professionals and commissioners/policymakers and not by stroke survivors or carers, emphasising the importance of treating vulnerable patients in a timely manner and provide proactive patient-centred care. This is in line with the NHS Long Term Plan set in 2019.[66] Patients/carers who participated in the focus groups were relatively mobile and maybe this was less of a priority for them.

These solutions, which are delivered through a DSS integrated with the EHR system and based on data from a linked population dataset, have the potential to be an instrument of change in clinical practice. This will be done by providing scientific evidence at the point of clinical care (eg, personalised treatments and their benefit based on the individual's risk factors), while simultaneously collecting information from that care (eg, treatments selected by the patient, desired outcomes, predicted stroke risk) to promote innovation in optimal healthcare delivery.[17]

### Strengths and limitations

Although the core focus of the DSS (prevention of a future stroke) was identified by patients as a priority, having a single focus might hinder discussions of other important problems (eg, depression, social isolation). Such issues may even have a larger perceived impact on long-term outcomes after stroke, for example, improving

mental health or access to social care services, which were also brought up by stakeholders as a priority to address long-term care for stroke survivors with multimorbidity,[14] and were raised as a concern in the usability and acceptability evaluations. Depression is indeed a risk factor of stroke,[67] and the treatment 'manage low mood/depression' will be displayed to all patients, enabling healthcare professionals to relate to this aspect and propose ways to manage this (eg, medication, referral to a professional, group therapy).

In a study assessing stroke survivors' self-reported needs,[68] >50% of long-term stroke survivors reported an unmet need for stroke information (eg, cause, prevention of recurrence). The proposed decision aid offers a meaningful starting point for addressing this common unmet need. Evidence suggests that the provision of lifestyle advice from healthcare professionals' is effective in changing health behaviours[69] and healthcare professionals' communication is positively correlated with patient adherence to treatments.[70] However, a conversation-based DSS also relies on the attitudes and communication skills of the healthcare professionals, which have been found to vary.[71] Interactive SDM skill training has improved SDM skills and promoted positive attitudes.[72] Training healthcare professionals in communication skills for SDM has also been shown to result in substantial and significant improvement in patient adherence to treatments.[70] Hence, interactive SDM skills training workshops will have to complement the use of the DSS. Patients are also likely to need support and preparation with taking part in SDM during the consultation.[72]

The design of DOTT meets the IPDAS collaboration criteria for quality decision aids.[23] Specifically, DOTT was designed to incorporate principles of SDM, by presenting stroke survivors with information about their treatment options and likely outcomes, presenting the risks and benefits of each option, and engaging the healthcare professional and stroke survivor in a joint conversation about the patient's preferences.[32] Furthermore, DOTT evolves from a systematic development process, uses non-technical language and presents information in a balanced manner that allows for comparisons across alternatives.[23] Wearable sensors (eg, Fitbit, Apple Watch, blood pressure monitor) could further help patients monitor and self-manage the selected treatments (eg, control blood pressure, increase physical activity) outside the consultation .[73] In the future, data from wearable sensors could be integrated to the EHR, and DOTT could use this information to improve its risk prediction model and treatment recommendations.

In the usability and acceptability evaluation, stroke survivors and GPs found DOTT to be both useful and usable. GPs perceived that the decision aid helped with structuring the consultation and eliciting patients' preferences for treatments. Stroke survivors felt it provides a good way to understand the different treatment options and select the ones that best suits their preferences. GPs' main concern was that the decision aid would increase consultation times. Indeed, time constrains were identified as the main barrier for the adoption of innovations by family physicians.[74 75] A possible solution could be to use the decision aid as part of a clinical review after stroke, which is usually longer (eg, 3 months, 6 months and annual review) and by dedicated healthcare professionals which are less limited in time such as stroke nurses and pharmacists working in GPs' practices that are trained to consult patients with chronic and long-term health conditions.

## CONCLUSION

Engaging various stakeholders throughout the design and evaluation process ensures that the intervention (features and functions) is in line with the needs reported by the different stakeholders (eg, stroke survivors, healthcare professionals, policymakers). DOTT has demonstrated the potential to reduce stroke recurrence by adopting a data-driven user-centred approach. DOTT urges clinicians to shift away from the professionally led advice-giving approach typically used in medical consultations to one which collaboratively and actively engages the patient in decision-making and respects patient choice and autonomy. This may lead to stroke survivors taking ownership for the treatment decisions, improving their adherence to the agreed management plan and thus reducing their stroke risk. A future feasibility study and subsequent clinical trial will evaluate the effectiveness of DOTT in improving decision-making quality, and whether it affects risk factor levels and risk of recurrence. While DOTT currently targets stroke risk factors only, the design approach and its features could be used for a range of chronic diseases requiring long-term management, paving the way to a set of standards for delivering LHS interventions in clinical practice.

**Acknowledgements** The research team are very grateful to all the stakeholders that participated in the study including stroke survivors, carers, healthcare professionals, policymakers and commissioners. We would like to thank specifically the King's College London Stroke Research Patients and Family Group for their valuable comments that helped improve the design of the decision aid.

**Contributors** Conception and design of study: TP, IJM, ES, MAV, CK, CDAW and VC. Data collection: TP, IJM and ES. Thematic analysis and interpretation of data: TP, IJM, ES, MAV and VC. Initial draft of manuscript: TP. Revising the paper critically for important intellectual content: TP, IJM, ES, MAV, CM, CDAW and VC. Sign-off final version of manuscript: TP, IJM, ES, MAV, CM, CDAW and VC.

**Funding** This work was supported by the National Institute for Health Research (NIHR) Collaboration for Leadership in Applied Health Research and Care (CLAHRC) South London at King's College Hospital NHS Foundation Trust, and the NIHR Biomedical Research Centre, Guy's and St Thomas' NHS Foundation Trust and King's College London, UK (award number NIHR CLAHRC-2013-10022). Porat and Curcin were also supported by the EPSRC CONSULT grant (EP/P010105/1).

**Competing interests** None declared.

**Patient consent for publication** Not required.

**Provenance and peer review** Not commissioned; externally peer reviewed.

**Data sharing statement** All data relevant to the study are included in the article or uploaded as supplementary information.

**Author note** Checklist for reporting guidelines: the authors used SRQR guidelines for reporting qualitative research.

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
