## [Reviewer comments · BMJ Open]

ARTICLE DETAILS

TITLE (PROVISIONAL)	Collaborative design of a decision aid for stroke survivors with multimorbidity: a qualitative study in the UK engaging key stakeholders
AUTHORS	Porat, Talya; Marshall, Iain; Sadler, Euan; Vadillo, Miguel; McKeivitt, Christopher; Wolfe, Charles; Curcin, Vasa

VERSION 1 – REVIEW

REVIEWER	Ian Kronish Center for Behavioral Cardiovascular Health Columbia University Irving Medical Center United States
REVIEW RETURNED	18-Apr-2019

GENERAL COMMENTS	The manuscript was clearly written and succinctly described the development of a decision aid for stroke survivors. Strengths included the use of qualitative methods to inform the design of the tool, engagement of a diverse group of stakeholders, as well as the use of user centered design to iteratively refine the tool. There were also some limitations, such as how to overcome challenges of implementing the tool in clinical practice. Some more specific comments that may be helpful for improving the manuscript are provided below: 1. The focus groups focused on identifying stakeholder priorities for information - yet other barriers than lack of information are likely to underlie low adherence to risk reducing health behaviors. Could the authors expand on why/how they selected information needs as a focus for these focus groups?2. The recruitment and sampling strategy were not clearly specified. Particularly with respect to stroke survivors, but also for providers, there was concern that the participants enlisted into the focus groups may not have been typical of stroke survivors more broadly, particularly those stroke survivors with poor risk factor control that may have the most to benefit from a shared decision making aid. Did the authors do anything to ensure they had a representative group of stroke survivors?3. Could the authors clarify whether the study team had decided to develop a decision aid prior to collecting their qualitative data, or if this emerged organically after conducting the focus groups? Either way, it would be helpful if they made it clear how a decision aid became a central focus of their approach.4. It wasn't clear which version of the decision aid was tested during
--

	usability testing. Did it incorporate actual patient data in real time? Did it have comprehensive lists of personalised treatments and examples of stroke survivors' concerns? It wasn't clear how much of the work of the tool was completed or still a work-in-progress. 5. During usability testing, there was mention on p.11 that stroke survivors would like to use the tool frequently. Was this a planned use of the tool? 6. In the Discussion, the authors mention wearable sensors being integrated into the tool - did this emerge from their qualitative data? Could they elaborate on this further? 7. The data visualization of risk was innovative. Do the authors plan or recommend comparing their innovative visualization with more traditional risk communication data visualizations? 8. Could the authors comment on the feasibility of some of the technological components including those that aim to be embedded in the EHR (e.g., sensors) and to be continuously updated with data from the health system over time (i.e., learning health system approach). Were these components aspirational or were they ready to be implemented and how? 9. How did the authors settle on a follow-up interval of 3 months? That sounds rather long for someone with a recent stroke. 10. The authors describe adding emotional and mental health components to their decision aid in response to concerns by stroke survivors and physicians. Can they elaborate on how they did this? 11. Could the authors provide additional details about the costs and feasibility of their approach.
--	---

REVIEWER	Brodie Sakakibara University of British Columbia Canada
REVIEW RETURNED	29-Apr-2019

GENERAL COMMENTS	Thank you for the opportunity to review this paper, reporting on the development and usability of a clinical decision support system (Deciding on Treatments Together, DOTT) for the secondary prevention of stroke. The strengths of the paper, include the focus on a very important issue (i.e. secondary stroke prevention where there are few available and effective resource), as well as a very systematic approach to the development of the program, incorporating the input from a wide range of stakeholders. While this paper is well-written and has many strengths, there are also several conceptual and methodological issues, noted below. A main concern that I'll highlight here (as well as in the below), is the uncertainty of the metrics used in the DOTT to indicate stroke risk, reduction in stroke risk, and how treatment effects are calculated to reduce stroke risk. Page 3, line 32: Suggest to expand and clarify, why long term stroke care is complex? What are the complexities?
---

	Page 3, line 34: More details of LHS and DSS are needed. First what are LHS and DSS, and what is the evidence supporting their use? Page 4, line 57: Please clarify the rationale of examining 'multimorbidity'. I think this is a very important issues among people with stroke, and needs further emphasis. Page 4, line 59: "All participants signed a consent form," is redundant with information in the above paragraph and may be removed here. Page 4, line 59: Please clarify how the 4 ideas for interventions that were discussed in the focus groups and interviews were determined. Please also provide more details on questions that the groups focused on addressing. Page 5, line 5: '...core stakeholder group.' Please clarify the core stakeholder group. What is this group considered 'core'? Page 5, line 7: It is not clear of the difference between this manuscript which focuses on the collaborative development of the DSS, and the authors' statement that, "Full details of the method for this stage have been published elsewhere." Page 5, line 11: The International Patient Decision Aids Standards and SDM model for clinical practice require further details. For example, what are they, how have they been used, how are they used in this research? Page 5, line 40: Further details of the System Usability Scale are required. For example, what does this questionnaire measure? how many items? any psychometric properties? Page 6, line 15: it is not clear why the usability evaluations were audio recorded? Please clarify the reason for recording these sessions. Page 6, line 28: Please clarify the statement in brackets (requirement from a DSS). Page 6, line 28: It is not clear which of the 'four ideas for interventions' (page 4, line 59) these themes relate to. Please clarify how the themes related to the 'four ideas'. Page 9, line 41: Please clarify how the development of DOTT aligns with the 4 ideas from Page 4, line 59. Page 9, line 59: Please clarify and provide more detail on how 'predicted stroke risk' will be calculated. Is this a validated measure of stroke risk? Page 10, line 17: Please clarify how the relative risk reductions for each of the recommended treatments will be calculated. Page 10, line 39. Please clarify why 3 months follow-up was chosen. Additionally, please clarify somewhere in the paper, the 'time post
--	--

	stroke', when people should begin using the DOTT. Page 10, line 47: Again, please clarify the how recurrent stroke risk will be calculated, and what determines 'high risk'? Page 10, line 56: Please provide more detail on the LHS. Page 11, line 22: Is the decision aid the DOTT (a DSS)? If yes, for consistency, suggest to use DSS throughout the manuscript. Page 11, line 52: Please clarify how the treatment effects are calculated and how they are put on the same scale as the stroke risk measure. Page 12, line 19: Please clarify the learning system. That is, what is it? Page 12, line 27: Please clarify what behaviours will be changed. Also please clarify the reasoning as to why the decision aid may motivate people to change behaviour in light of current evidence that it is difficult to change behaviour in stroke patients, as per the Cochrane review cited in the introduction. Page 12; line 54: Please clarify why only 10 minutes is allotted for the consultation. It would seem like much more time would be need to go through all the components of the DOTT. Pge 13, line 23: Please expand on the themes and solutions that have been used to support other patient groups, and used in DOTT. Also identify whether the solutions worked in these groups. By doing so will emphasize the 'evidence-based' aspect of DOTT. Page 14, line 29: It is not clear where education of causes and prevention of stroke is included in the DOTT, especially if the consultations are only 10 minutes long. Please clarify.
--	--

REVIEWER	Ryc Aquino University of Cambridge, UK
REVIEW RETURNED	30-Apr-2019

GENERAL COMMENTS	Dear authors, Thank you for the opportunity to review this work, which focussed on a qualitative study involving engaging with multiple stakeholders in the design and evaluation of an intervention for the improvement of risk factors management and secondary stroke prevention in primary care. This is an important and timely piece of work, which could inform future research and has the potential to improve patient/service outcomes. Below are some comments, which I hope you will find useful: The introduction is clear and well-written. To clearly signpost the reader to the study objectives, the authors might wish to add a subheading 'aims and objectives' (para. 2 of page 4) and outline these. Concerning the methods, please include focus group/interview topic guides for each stage as supplementary files, and within the main text provide a brief summary of the topics explored. It would also be useful to see a study flow/figure to depict the three stages of inquiry.
---

	With regard to the eligibility criteria for participation, were there any restrictions on time since stroke? If so please include some information on this. Perhaps the authors could also include stroke survivors' characteristics (e.g. no. years post-stroke, age) to provide some context to the participants similar to how you described different health and social care professional roles in Table 1. Concerning stage 2, 'collaborative design and prototyping...', it is unclear how the data/feedback gathered from the SRPFG after the fourth focus group was fed into the final focus group/the rest of the intervention. I would suggest providing a diagrammatic summary of the intervention development phases and revisions, including whose data/feedback fed into which phase to clearly see the lifecycle of the DSS. Also, for brevity just state that 22 members of the SRPFG were involved in the presentation/meeting to discuss the DSS Besides the questionnaires for stage 3, 'usability and acceptability...', were any other qualitative data collected to explore usability and acceptability? If not could this be explored in the discussion? The authors might also wish to consider providing their working definitions of usability and acceptability. The authors report conducting a thematic analysis (TA) – please could you provide further detail on this analysis, specifically: the rationale for a TA; what the roles of the analysts were (e.g. were they also the moderators/facilitators in the focus groups, disciplinary backgrounds -- this will then address #6 of your SRQR checklist); how the analysis was conducted in light of the 3-stage study, whether the analysis was data or theory-driven, and whether data from main study participants were analysed together with or separately from the SRPFG data? Regarding the results: The themes presented were meaningful and relevant to the development of the DSS. In particular, the themes related to risk communication and how such information is presented and then addressed. I was very interested to read theme 3, 'compare stroke survivors' perceived stroke risk...' – this looks highly relevant and I am wondering whether how this compared with the rest of the focus groups/interviews as the paragraph currently reads as if it focussed on the fourth group alone. If this theme did not come up in other groups/interviews, it would be interesting to discuss this in the context of the wider findings. Theme 8 'identify stroke survivors...' is also interesting and relevant for the development of the DSS particularly for care providers/commissioners – I wonder whether any stroke survivors/carers had any opinions concerning this? The authors might wish to consider exploring this in the context of the wider findings. The usability and acceptability evaluation section of the paper could be clarified. At the moment there are 8 further themes within this section split into two: identified themes (5 themes) and concerns (2 themes). This section would be clearer if the themes are consolidated to highlight the key messages: 1) the value/importance of the DSS (encompassing themes 1+4+5), 2) the functionality of the DSS (themes 2+3), and 3) concerns about the DSS (themes 1+2 – Concerns subsection).
--	---

	The discussion is clearly written and consider the relevant literature. In relation to the points about risk communication and health decision-making, the authors might wish to consider relevant theories (e.g. Glanz et al.; Prochaska et al.) to inform this discussion. Overall, the authors present a robust piece of research that demonstrates the use of collaborative approaches to designing decision aids for stroke survivors and I would support its publication following revisions.
--	---

VERSION 1 – AUTHOR RESPONSE

#	Issue	Response
1	Reviewer 1: The focus groups focused on identifying stakeholder priorities for information - yet other barriers than lack of information are likely to underlie low adherence to risk reducing health behaviors. Could the authors expand on why/how they selected information needs as a focus for these focus groups?	Thank you for your comments. The aim of the study was to engage stakeholders in identifying needs and design data-driven solutions (digital interventions), that will be part of a Learning Health System, therefore we focused on data/information needs.
2	Reviewer 1: The recruitment and sampling strategy were not clearly specified. Particularly with respect to stroke survivors, but also for providers, there was concern that the participants enlisted into the focus groups may not have been typical of stroke survivors more broadly, particularly those stroke survivors with poor risk factor control that may have the most to benefit from a shared decision making aid. Did the authors do anything to ensure they had a representative group of stroke survivors?	This information was added to the ‘Method’ section under ‘Data collection’ (p.5). “Stakeholders were purposively sampled to include stroke survivors (i.e. men and women, with a range of disabilities and long-term conditions, risk factors and length of time since their stroke) and professionals providing all types of stroke care and support.” A limitation which we noted in the paper (‘Strengths and limitations of this study’) is the exclusion of stroke survivors who could not attend the focus groups (e.g., less mobile or housebound). It is important to note (and was added to the text) that transport was arranged for less mobile patients (p.5).
3	Reviewer 1: Could the authors clarify whether the study team had decided to develop a decision aid prior to collecting their qualitative data, or if this emerged organically after conducting the focus groups? Either way, it would be helpful if they made it clear how a decision aid became a central focus of their approach.	The idea of a decision aid to target multiple risk factors and improve secondary stroke prevention was an outcome of the first stage. We clarified this in the text, under Method/Data collection/Stage 1 (p. 5). “Targeting multiple risk factors after stroke was identified among stakeholders as a key priority, and a DSS to improve secondary prevention after stroke to target multiple risk factors was subsequently chosen within a smaller core stakeholder group (FG3) for further development.”

		In addition, we added a diagrammatic summary of the development of DOTT, including the data that fed the different stages and the outputs (Figure 1).
#	Issue	Response
4	Reviewer 1: It wasn't clear which version of the decision aid was tested during usability testing. Did it incorporate actual patient data in real time? Did it have comprehensive lists of personalised treatments and examples of stroke survivors' concerns? It wasn't clear how much of the work of the tool was completed or still a work-in-progress.	We added the version of the decision aid that was used for the evaluation and its functionality in Method/Data collection/Stage 3 (p. 6). For the usability evaluation, DOTT prototype had the following functionality and flow:  • Stroke survivors (patients) indicated their perceived risk of having a recurrent stroke. • GPs entered the patient's characteristics (age, gender, clinical conditions). • The system displayed a 'typical' recurrent stroke risk (age group specific average)³⁷ and the most effective treatments based on the patient's characteristics. • The benefit of each treatment in terms of reducing the stroke risk was displayed. Estimated relative stroke risk reductions were calculated based on the existing literature.³⁸⁻⁴¹ • Information and common concerns for each treatment were displayed. • The GP and patient decided on a management plan whilst identifying desired clinical and patient outcomes. • Patients were told that their management plan would be printed to take home.
5	Reviewer 1: During usability testing, there was mention on p.11 that stroke survivors would like to use the tool frequently. Was this a planned use of the tool?	The aim is to use the decision aid in each consultation in order to review the management plan and decide together (GP and patient) whether to add, modify or remove treatments. This is mentioned in the section Results/Development of DOTT decision support system/item 7 (p.11).
6	Reviewer 1: In the Discussion, the authors mention wearable sensors being integrated into the tool - did this emerge from their qualitative data? Could they elaborate on this further?	Wearable sensors did not emerge from the qualitative data. This is a direction we are thinking of in order to support patients outside the consultation in adhering to the treatments they selected during the consultation using DOTT and in improving our prediction model. This was elaborated in the discussion (Discussion/Strengths and limitations, p.16).

		“Wearable sensors (e.g., Fitbit, Apple Watch, blood pressure monitor) could further help patients monitor and self-manage the selected treatments (e.g., control blood pressure, increase physical activity) outside the consultation. In the future, data from wearable sensors could be integrated to the EHR, and DOTT could use this information to improve its risk prediction model and treatment recommendations.”
#	Issue	Response
7	Reviewer 1: The data visualization of risk was innovative. Do the authors plan or recommend comparing their innovative visualization with more traditional risk communication data visualizations?	Yes, we are currently designing this study – comparing our data visualisation to the traditional risk visualisation (cates plot).
8	Reviewer 1: Could the authors comment on the feasibility of some of the technological components including those that aim to be embedded in the EHR (e.g., sensors) and to be continuously updated with data from the health system over time (i.e., learning health system approach). Were these components aspirational or were they ready to be implemented and how?	DOTT is a working prototype. As mentioned in the ‘Conclusion’, we are currently working on the development of DOTT for a feasibility study (including integrating it to the EHR and adding the LHS components). Integrating information from sensors is a future component, and will not be included in the forthcoming feasibility study.
9	Reviewer 1: How did the authors settle on a follow-up interval of 3 months? That sounds rather long for someone with a recent stroke.	Current clinical guidelines (NICE guidelines) for ‘Secondary prevention following stroke and TIA’ recommend primary care follow up on discharge, at 6 months and then only annually. Based on this and what we thought was reasonable to expect from healthcare professionals and to provide enough time for patients to adhere to the selected treatments, we proposed intervals of three months. This interval will also be evaluated in the feasibility study and may be modified. This was clarified in Results/Development of DOTT decision aid/issue 7, p.11. “Current NICE guidelines⁴⁵ for ‘Secondary prevention following stroke and TIA’ recommend primary care follow up on discharge, six months and then annually. A three-month follow up was selected as a reasonable interval for healthcare professionals and to provide enough time for patients to adhere to the selected treatments.”
10	Reviewer 1: The authors describe adding emotional and mental health components to their decision aid in response to concerns by stroke survivors and	We added to the possible treatments ‘manage low mood/depression’. This clarification was added to the text in the Discussion/Strengths and limitations (p.15).

	physicians. Can they elaborate on how they did this?	“...the treatment ‘manage low mood/depression’ will be displayed to all patients, enabling healthcare professionals to relate to this aspect and propose ways to manage this (e.g., medication, referral to a professional, group therapy).
11	Reviewer 1: Could the authors provide additional details about the costs and feasibility of their approach.	We plan to evaluate main costs (e.g., development, installation, training, support) and other feasibility issues in the forthcoming feasibility study.
12	Reviewer 2: Page 3, line 32: Suggest to expand and clarify, why long term stroke care is complex? What are the complexities?	Thank you for your comments. The following text was added to clarify why long-term stroke care is complex (in the ‘Introduction’, p.3):
#	Issue	Response
		“Stroke survivors commonly experience multimorbidity.¹² Gallacher and colleagues found that 94% of the people with stroke had one or more additional morbidities and often experienced longterm physical, psychological and social consequences.¹² This makes improving long-term stroke care a complex endeavour, requiring patient engagement, high quality up-to-date information and a holistic approach which focuses on the patient and not on the disease.¹³”
13	Reviewer 2: Page 3, line 34: More details of LHS and DSS are needed. First what are LHS and DSS, and what is the evidence supporting their use?	More details on the LHS and DSS were added to the ‘Introduction’ (p.3). “The Learning Health System (LHS) ‘focusses on approaches to capture data from clinical encounters and other health-related events, analyse the data to generate new knowledge, and then apply this knowledge to continuously inform and improve health decision making and practice.’^{15(p.177)} In a recent report (2019) stating what the NHS can learn from the LHS, the authors argue that it is necessary to utilise data to transform services, not just to digitise current ways of working.¹⁶” “Decision support systems (DSS) which aim to analyse a patient’s characteristics to provide tailored recommendations (such as for diagnosis,¹⁸ treatment or long-term management), implement this transfer of evidence into practice. This is done particularly when used in conjunction with sources of ‘Real World Data’¹⁹ such as EHR systems that capture detailed data on specific conditions.” Evidence supporting DSS use to facilitate shared decision making is described on p.4 and evidence of the use of a LHS was added to the Introduction (p.3):

		“A few studies have reported that engaging stakeholders to develop a LHS and integrated DSS improved patient outcomes and processes of care for individuals with long-term conditions. ^{20,21} ”
14	Reviewer 2: Page 4, line 57: Please clarify the rationale of examining 'multimorbidity'. I think this is a very important issues among people with stroke, and needs further emphasis.	Please see our response to issue 12. We believe that the text addition emphasises the rationale for focusing on stroke survivors with multimorbidity.
15	Reviewer 2: Page 4, line 59: "All participants signed a consent form," is redundant with information in the above paragraph and may be removed here.	Thank you, was removed.
16	Reviewer 2: Page 4, line 59: Please clarify how the 4 ideas for interventions that	Data from focus groups and interviews were audio recorded, transcribed in full and stored in NVivo.
#	Issue	Response
	were discussed in the focus groups and interviews were determined.	Qualitative data were analysed using a thematic analysis approach where we identified the four priorities for information needs. To clarify this point we elaborated on the thematic analysis process (Method/Data analysis, p.7). “...This involved two authors (TP, ES) assigning codes and refining themes from the data, noting similarities and differences between stakeholder perspectives. The two authors have doctoral/postdoctoral experience in conducting and analysing qualitative research and stakeholder engagement approaches in applied health research.”
17	Reviewer 2: Page 5, line 5: '...core stakeholder group.' Please clarify the core stakeholder group. What is this group considered 'core'?	This was clarified in the text (Method/Data collection/Stage 1, p.5): “This core stakeholder group (N=12) comprised stroke survivors, healthcare professionals, carer, policy maker and commissioner, and worked collaboratively with the research team to subsequently design the intervention and to provide their active feedback.”
18	Reviewer 2: Page 5, line 7: It is not clear of the difference between this manuscript which focuses on the collaborative development of the DSS, and the authors' statement that, "Full details of the method for this stage have been published elsewhere."	We published a paper which focuses on the process of engaging stakeholders in the use of clinical and research data, based only on the first stage of the study - Exploring stakeholder priorities for data and information needs (Sadler et al., 2017).
19	Reviewer 2: Page 5, line 11: The International Patient Decision Aids Standards and SDM model for clinical practice require further details. For	Due to word limitations we cannot provide all the information, however we provided additional details on each one (in Method/Data collection/Stage 2, p.5).

	example, what are they, how have they been used, how are they used in this research?	“The initial design of the DSS to improve secondary stroke prevention and target multiple risk factors after stroke was informed by the first stage and guided by the International Patient Decision Aids Standards (IPDAS),²³ which provides a framework and standards for the design of patient decision aids, and the SDM model for clinical practice.³² The latter provides a model of how to conduct shared decision making in practice based on providing patients choice, a range of options and involving them in ‘decision talk’.” We also related to the Patient Decision Aids Standards in the Discussion/Strengths and limitations, p.16. “The design of DOTT meets the IPDAS collaboration criteria for quality decision aids.²³ Specifically, DOTT
#	Issue	Response
		was designed to incorporate principles of SDM, by presenting stroke survivors with information about their treatment options and likely outcomes, presenting the risks and benefits of each option, and engaging the healthcare professional and stroke survivor in a joint conversation about the patient’s preferences.³² Furthermore, DOTT evolves from a systematic development process, uses non-technical language and presents information in a balanced manner that allows for comparisons across alternatives.²³”
20	Reviewer 2: Page 5, line 40: Further details of the System Usability Scale are required. For example, what does this questionnaire measure? how many items? any psychometric properties?	We added this information in Method/Data collection/Stage 3, p.6) and included the System Usability Scale as a supplementary file. “The SUS is composed of 10 questions and has been shown to be a reliable and psychometrically validated tool.³⁶ Ratings were provided on 5-point Likert scales from 1 (strongly disagree) to 5 (strongly agree), with higher ratings indicating higher satisfaction.”
21	Reviewer 2: Page 6, line 15: it is not clear why the usability evaluations were audio recorded? Please clarify the reason for recording these sessions.	Only the interview after the usability evaluation was audio recorded, this was clarified in the text (Method/Data analysis, p.7). “Data from focus groups and interviews were audio recorded, transcribed in full and stored in NVivo (Version 11).”
22	Reviewer 2: Page 6, line 28: Please clarify the statement in brackets (requirement from a DSS).	This was clarified (Results/Focus groups and interviews, p.7): “Eight themes related to improving secondary prevention and management of multiple risk factors after stroke were identified from focus groups and interviews”

23	Reviewer 2: Page 6, line 28: It is not clear which of the 'four ideas for interventions' (page 4, line 59) these themes relate to. Please clarify how the themes related to the 'four ideas'.	The themes related only to the design of a decision support system to improve secondary stroke prevention. The selection of this intervention was the outcome of the first stage (in Method/Data collection/Stage 1, p.5): “Targeting multiple risk factors after stroke was identified among stakeholders as a key priority, and a DSS to improve secondary prevention after stroke to target multiple risk factors was subsequently chosen within a smaller core stakeholder group (FG3) for further development.” We also added a diagrammatic summary of the development of DOTT, including the data that fed the different stages and the outputs (Figure 1).
#	Issue	Response
24	Reviewer 2: Page 9, line 41: Please clarify how the development of DOTT aligns with the 4 ideas from Page 4, line 59.	It aligns only with the forth idea which was selected in the first stage. See answer above.
25	Reviewer 2: Page 9, line 59: Please clarify and provide more detail on how 'predicted stroke risk' will be calculated. Is this a validated measure of stroke risk?	The predicted cardiovascular/stroke risk calculator used in DOTT will be based on the South London Stroke Register (SLSR) and Lambeth Datanet (LDN) datasets, which will provide an estimated risk of the stroke survivor to have a recurrent stroke based on their risk factors (e.g., age, gender, hypertension, atrial fibrillation). The analytic model will also calculate the reduction of risk if one or more treatments are taken. This was mentioned in the Results section/Development of DOTT decision support system/issues 2 and 4 (p.11). The feasibility study will inform the sample size to validate our risk score for secondary prevention.
26	Reviewer 2: Page 10, line 17: Please clarify how the relative risk reductions for each of the recommended treatments will be calculated.	See response above (issue 25).
27	Reviewer 2: Page 10, line 39. Please clarify why 3 months follow-up was chosen.	Please see our response to issue 9.
28	Reviewer 2: Page 10, line 47: Again, please clarify the how recurrent stroke risk will be calculated, and what determines 'high risk'?	See response to issue 25. 'High risk' value/range (i.e., when to alert the practice/healthcare professional) is still need to be determined.
29	Reviewer 2: Page 10, line 56: Please provide more detail on the LHS.	Details on the LHS were added to the Introduction. See response to issue 13.

30	Reviewer 2: Page 11, line 22: Is the decision aid the DOTT (a DSS)? If yes, for consistency, suggest to use DSS throughout the manuscript.	A decision aid is a type of DSS that has been designed specifically to facilitate shared decision making, and hence we think this distinction is important. In addition, we used the 'International Patient Decision Aids Standards' to design DOTT and this might confuse the readers. We did however emphasise the distinction between the two in the Introduction (p.4) and under 'Development of DOTT decision aid' (p.10). "...and several DSS that have been designed to facilitate SDM during the consultation (i.e., decision aids) have shown..." (p.4) "DOTT is a computerised decision aid (i.e., a DSS designed to facilitate SDM), integrated..." (p.10)
31	Reviewer 2: Page 11, line 52: Please clarify how the treatment effects are calculated and how they are put on the same scale as the stroke risk measure.	See response to Issue 25. For the current DOTT prototype we used a 'typical' recurrent stroke risk (age group specific average). Estimated benefit for each treatment were calculated based on existing literature. This information was added to Method/ Data collection/Stage 3 (p.6).
#	Issue	Response
		"The system displayed a 'typical' recurrent stroke risk (age group specific average)³⁷ and the most effective treatments based on the patient's characteristics." "The benefit of each treatment in terms of reducing the stroke risk was displayed. Estimated relative stroke risk reductions were calculated based on the existing literature.³⁸⁻⁴¹"
32	Reviewer 2: Page 12, line 19: Please clarify the learning system. That is, what is it?	See response to issue 13.
33	Reviewer 2: Page 12, line 27: Please clarify what behaviours will be changed. Also please clarify the reasoning as to why the decision aid may motivate people to change behaviour in light of current evidence that it is difficult to change behaviour in stroke patients, as per the Cochrane review cited in the introduction.	The types of behaviours were clarified in the text (Results/Usability and acceptability evaluation/Identified themes, p. 13). "All GPs and stroke survivors (N=16) believed that the decision aid could motivate patients to change behaviour (e.g., take their medication to reduce blood pressure, increase physical activity, eat healthy)." Stakeholders argued that sharing decisions with the patients may increase their feeling of ownership over their health and hence their adherence to the selected treatments. This was clarified in the text (p.13).

		“A number of GPs and stroke survivors agreed that sharing decisions and enabling patients to select the treatments that best meet their preferences and goals, may increase patients’ feeling of ownership over their health and improve adherence to the selected treatments.”
34	Reviewer 2: Page 12; line 54: Please clarify why only 10 minutes is allotted for the consultation. It would seem like much more time would be need to go through all the components of the DOTT.	In the UK, a standard clinical consultation in primary care (with the GP) is allotted 10 minutes. GPs related to this limitation.
35	Reviewer 2: Page 13, line 23: Please expand on the themes and solutions that have been used to support other patient groups, and used in DOTT. Also identify whether the solutions worked in these groups. By doing so will emphasize the 'evidence-based' aspect of DOTT.	As mentioned in the Introduction (p.4), decision aids used during the clinical consultation have shown improved treatment adherence and clinical outcomes. The themes and solutions that have been used to support other patient groups, and used in DOTT were found useful and therefore are recommended in design of decision aids (e.g., are part of the International Patient Decision Aid Standards (IPDAS)). This was clarified in the Discussion (p.14). “These themes were found useful and are recommended in SDM tools (e.g., in the IPDAS²³).”
36	Reviewer 2: Page 14, line 29: It is not clear where education of causes and prevention	DOTT decision aid includes information on common patients’ concerns relating to the different
#	Issue	Response
	of stroke is included in the DOTT, especially if the consultations are only 10 minutes long. Please clarify.	treatments (similar to Q&A), which will aid in identifying and addressing barriers to treatment adherence and eliciting preferences. This is described in Results/Development of DOTT decision aid’/Issue 5, p.11.
37	Reviewer 3: The introduction is clear and well-written. To clearly signpost the reader to the study objectives, the authors might wish to add a subheading ‘aims and objectives’ (para. 2 of page 4) and outline these.	Done (Introduction/Aims and objectives, p.4). “The aim of this study was to engage key stakeholders to identify priorities and information needs in long term stroke care and collaboratively design and evaluate a selected intervention that could be integrated as part of the EHR system informed by a LHS approach.”
38	Reviewer 3: Concerning the methods, please include focus group/interview topic guides for each stage as supplementary files, and within the main text provide a	We have now included the focus group/i nterview topic guides as supplementary files, added a and brief summary in the main text /Data (Method collection/Stage 1, p.5).

	brief summary of the topics explored. It would also be useful to see a study flow/figure to depict the three stages of inquiry.	“In brief, in the initial engagement meeting participants were introduced to the concept and then in three separate focus groups (user/carer; health and social care professionals; commissioners and policy makers) they were asked to identify priorities and potential solutions that may be derived from the clinical data to improve long-term stroke care for stroke survivors with multimorbidity. Then, in the larger group process of priority setting and consensus facilitator (ES), stakeholders identified a priorities and solutions to improve long-term management of stroke (i.e. improving co care; improving management of mental consequences; better access to health care; and targeting multiple risk factors). Multiple risk factors after stroke was identified among stakeholders as a key priority, an improve secondary prevention after stroke multiple risk factors was subsequently chosen a smaller core stakeholder group (FG3) for development. This core stakeholder group comprised stroke survivors, healthcare professional, carer, policy maker and commissioner, a collaboratively with the research team to subsequently design the intervention and to provide their active feedback .” In addition, based on your recommendation (40) , we added a diagrammatic summary of the development of DOTT, including the data that fed the different stages and the outputs (Figure 1).
#	Issue	Response
39	Reviewer 3: With regard to the eligibility criteria for participation, were there any restrictions on time since stroke? If so please include some information on this. Perhaps the authors could also include stroke survivors’ characteristics (e.g. no. years post-stroke, age) to provide some context to the participants similar to how you described different health and social care professional roles in Table 1.	There were no restrictions on time since stroke. Stroke survivors were purposively sampled to include men and women with a range of disabilities, long-term conditions, risk factors and length of time since their stroke and this was added to the Method/Data collection, p.5.

40	Reviewer 3: Concerning stage 2, 'collaborative design and prototyping...', it is unclear how the data/feedback gathered from the SRPFG after the fourth focus group was fed into the final focus group/the rest of the intervention. I would suggest providing a diagrammatic summary of the intervention development phases and revisions, including whose data/feedback fed into which phase to clearly see the lifecycle of the DSS. Also, for brevity just state that 22 members of the SRPFG were involved in the presentation/meeting to discuss the DSS	Thank you for this helpful comment. A diagrammatic summary of the development of DOTT, including the data that fed the different phases and the outputs was added for clarification (Figure 1, p.5). We stated that 22 members were involved (we moved the information on the SRPFG to the 'Patient and public involvement' section under 'Method').
41	Reviewer 3: Besides the questionnaires for stage 3, 'usability and acceptability...', were any other qualitative data collected to explore usability and acceptability? If not could this be explored in the discussion? The authors might also wish to consider providing their working definitions of usability and acceptability.	In addition to the usability and acceptability questionnaires, we interviewed the participants (semi-structured interviews) asking them to provide feedback on the DSS. We added the interview questions to the supplementary files, and elaborated on this in Method/Data collection/Stage 3, p.6. "GPs and stroke survivors were interviewed after the simulated consultation, asking them to provide feedback on the DSS, including its strengths, limitations and suggestions for improvements." The terms 'usability' and 'acceptability' were defined (Method/Data collection/Stage 3, p.6). "Acceptability relates to the comprehensibility of the components of the decision aid, including its length, pace, amount of information, balance in presentation and overall suitability.³³ Usability is 'the extent to which a product can be used by specified users to achieve specified goals with effectiveness, efficiency, and satisfaction in a specified context of use'.³⁵"
42	Reviewer 3: The authors report conducting a thematic analysis (TA) – please could you provide further detail on this analysis, specifically: the rationale for a TA; what the roles of the analysts were (e.g. were	This information was added to Method/Data analysis, p.7. "...This involved two authors (TP, ES) assigning codes and refining themes from the data, noting
#	Issue	Response
	they also the moderators/facilitators in the focus groups, disciplinary backgrounds -- this will then address #6 of your SRQR checklist); how the analysis was conducted in light of the 3-stage study, whether the analysis was data or theory-driven, and whether data from main study participants were analysed together with or separately from the SRPFG data?	similarities and differences between stakeholder perspectives. The two authors have doctoral/postdoctoral experience in conducting and analysing qualitative data in applied health research."

43	Reviewer 3: Regarding the results: The themes presented were meaningful and relevant to the development of the DSS. In particular, the themes related to risk communication and how such information is presented and then addressed. I was very interested to read theme 3, ‘compare stroke survivors’ perceived stroke risk...’ – this looks highly relevant and I am wondering whether how this compared with the rest of the focus groups/interviews as the paragraph currently reads as if it focussed on the fourth group alone. If this theme did not come up in other groups/interviews, it would be interesting to discuss this in the context of the wider findings.	Thank you for this comment. We added a section about perceived risk in the Discussion (section 2, p.14-15). (2) Compare patient’s perceived risk with their predicted risk. This is a novel requirement from a DSS, which to our knowledge does not exist in current systems. Perceived risk of adverse outcomes such as stroke may be an important concept in understanding patient’s adherence to medication and recommended health behaviours.⁵⁸ Overall, patients tend to underestimate their own risk.⁵⁹ This tendency was also found when patients estimated their cardiovascular risk.⁶⁰ Weinstein refers to this underestimation as an “optimistic bias”.⁵⁹ For example, a recent study found that people with undiagnosed diabetes or prediabetes considerably underestimated their probability to have or develop diabetes.⁶¹ Lower perceived risk has been associated with poorer adherence to recommended health behaviours⁶² and hence a more realistic perception of risk may increase patients’ interest in risk reduction.⁶² Research has shown that individualised risk feedback was effective in increasing perceived stroke risk among patients who had underestimated their stroke risk at baseline.⁶³ This may imply that eliciting patients’ perceived risk and showing them the actual predicted risk can change their inaccurate risk perception and increase their interest in risk reduction.”
44	Reviewer 3: Theme 8 ‘identify stroke survivors...’, is also interesting and relevant for the development of the DSS particularly for care providers/commissioners – I wonder whether any stroke survivors/carers had any opinions concerning this? The authors might wish to consider exploring this in the context of the wider findings.	This theme was identified and prioritised by healthcare professionals and commissioners/policy makers, not patients/carers. We have elaborated on this in the Discussion, issue 4, p.15. “This theme was identified and prioritised by healthcare professionals and commissioners/policy makers and not by stroke survivors or carers, emphasising the importance of treating vulnerable patients in a timely manner and provide proactive patient-centred care. This is in line with the NHS Long Term Plan set in 2019.⁶⁶ Patients/carers who participated in the focus groups were relatively
#	Issue	Response
		mobile and maybe this was less of a priority for them.”

45	Reviewer 3: The usability and acceptability evaluation section of the paper could be clarified. At the moment there are 8 further themes within this section split into two: identified themes (5 themes) and concerns (2 themes). This section would be clearer if the themes are consolidated to highlight the key messages: 1) the value/importance of the DSS (encompassing themes 1+4+5), 2) the functionality of the DSS (themes 2+3), and 3) concerns about the DSS (themes 1+2 – Concerns subsection).	Thank you for this good suggestion, which we have now done (Results/Usability and acceptability evaluation/Identified themes, p.12-14).
46	Reviewer 3: The discussion is clearly written and consider the relevant literature. In relation to the points about risk communication and health decisionmaking, the authors might wish to consider relevant theories (e.g. Glanz et al.; Prochaska et al.) to inform this discussion.	Thank you. We have added relevant literature relating to perceived risk and health decision making in the discussion (Issue 2, p.14-15).

VERSION 2 – REVIEW

REVIEWER	Ian Kronish Columbia University Irving Medical Center United States
REVIEW RETURNED	14-Jun-2019

GENERAL COMMENTS	The authors satisfactorily addressed the prior comments of the Reviewers.
---

REVIEWER	Brodie Sakakibaa University of British Columbia, Canada
REVIEW RETURNED	24-Jun-2019

GENERAL COMMENTS	Thank you for the opportunity to again review this paper. It is much improved, and I thank the authors for their thoughtful revisions. I have one outstanding comment/concern that I feel was not adequately addressed neither in the responses nor in the paper, that is: Page 11, line 11, how is the predicted stroke risk calculated? Given that the intervention focuses on stroke prevention, the stroke risk calculation is a key component of the program. The authors state the risk score will be ‘calculated based on the patient’s information from the EHR and on rules generate from the linked dataset’ but do not specify what the ‘rules’ are or what variables will be used to calculate stroke risk. Please clarify how the stroke risk will be calculated.
--

VERSION 2 – AUTHOR RESPONSE

2	Reviewer 2: I have one outstanding comment/concern that I feel was not adequately addressed neither in the responses nor in the paper, that is: Page 11, line 11, how is the predicted stroke risk calculated? Given that the intervention focuses on stroke prevention, the stroke risk calculation is a key component of the program. The authors state the risk score will be 'calculated based on the patient's information from the EHR and on rules generate from the linked dataset' but do not specify what the 'rules' are or what variables will be used to calculate stroke risk. Please clarify how the stroke risk will be calculated.	As mentioned on p.6, for the usability evaluation, the system displayed a 'typical' recurrent stroke risk based on age³⁷. The final personalised risk model is under development and will include variables such as age, gender, medical history (e.g., hypertension, atrial fibrillation), type of stroke and time since stroke. This was clarified on p.11 (section 2).
---	--	---